**Data Availability Statement:** All files are available at https://osf.io/k3hfm/.

**Funding:** The funders had no role in study design, data collection and analysis, decision to publish, or

# The Theory of Planned Behavior during the COVID-19 pandemic: A comparison of health behaviors between Belgian and French residents

**Robin Wollast** [ID]*, **Mathias Schmitz, Alix Bigot, Olivier Luminet**

Research Institute for Psychological Sciences, UCLouvain, Ottignies-Louvain-la-Neuve, Belgium

\* Robin.wollast@hotmail.com

## Abstract

The COVID-19 pandemic presents a global crisis and authorities have encouraged the population to promote preventive health behaviors to slow the spread of the virus. While the literature on psychological factors influencing health behaviors during the COVID-19 is flourishing, there is a lack of cross-national research focusing on multiple health behaviors. The present study overcomes this limitation and affords a validation of the Theory of Planned Behavior (TPB) as a conceptual framework for explaining the adoption of handwashing and limitation of social contacts, two health behaviors that highly differ in their nature. Specifically, we compare TPB model on these two protective behaviors among people living in Belgium ($N = 3744$) and France ($N = 1060$) during the COVID-19 sanitary crisis. Data were collected from March 18 until April 19, 2020, which corresponds to the spring lockdown and the first peak of the pandemic in these countries. Results indicated that more positive attitudes, greater social norms, increased perceived control and higher intentions were related to higher adherence to handwashing and limitation of social contacts, for both Belgian and French residents. Ultimately, we argued that the TPB model tends to manifest similarly across countries in explaining health behaviors, when comparing handwashing and limitation of social contacts among individuals living in different national contexts.

## Introduction

In late December 2019, many hospitals in Wuhan, a city of 11 million people in Eastern China, reported a previously-unknown virus behind a number of pneumonia cases. Soon after, a seafood and live animal market was found to be associated with all cases and was identified as the source of the COVID-19 outbreak. The bulk of new cases being recorded each day are now outside China, and the virus has spread at some speed across the entire world (for a review of the COVID-19, see [1]).

Sanitary behaviors are crucial for slowing the spread of infectious diseases, especially when vaccines are not yet available. In this context, many countries followed public health

preparation of the manuscript. Mathias Schmitz and Robin Wollast received a salary from the Louvain Foundation.

**Competing interests:** The authors have declared that no competing interests exist.

recommendations [2] and reinforced health behaviors such as handwashing and limitation of social contacts to prevent the virus from further spreading. Behavioral changes as a response to these public health recommendations are heavily dependent on the acquisition of new social norms of interaction [3]. Social contacts patterns and hygiene behaviors are ingrained and are difficult to modify. In this context, the COVID-19 pandemic is an ideal testing ground to better understand how different factors can facilitate or hinder the adoption of these behaviors. Hence, the purpose of the present study is to investigate the adherence of these two major health behaviors recommended to face COVID-19, in the light of the Theory of Planned Behavior, among people living in Belgium ($N = 3744$) and in France ($N = 1060$). Understanding the mechanisms that improve the adoption and maintenance of health behaviors is critical to devising preventive interventions and new policies, as well as developing sustainable health emergency preparedness to deal with the current and future pandemics.

## Health behaviors during the COVID-19 pandemic

The present work focuses on two key health behaviors: Handwashing and limitation of social contacts. These two behaviors can be distinguished on many facets. Frequent handwashing is a classical preventive health behavior [4]. In a systematic review of physical interventions employed to reduce the transmission of respiratory viruses, handwashing was indicated to be effective with a meta-analytic summary estimate of a 45–55% reduction in transmission [5]. Similarly, a systematic review of the effectiveness of personal protective measures in preventing H1N1 pandemic influenza transmission in human populations indicated a 38% reduction in transmission with handwashing [6]. In sum, most people understand the importance of handwashing to reduce disease transmission and they already wash their hands frequently [7].

The second targeted behavior—limiting social contacts by staying home, avoiding public spaces and social distancing—is an avoidant behavior [8]. Avoidant behaviors are more challenging to adopt and maintain because they are counterintuitive and unnatural. While limiting social contacts has been found to be an effective strategy to flatten the epidemic curve, it is difficult for people to inhibit their natural tendency to interact with others [9]. Adopting such radical behavior change within a few days and for a long period of time is hard as compared to handwashing which was already considered as a habit in the population. Specifically, these social distancing rules are vital but they also undermine deep-rooted needs for togetherness and can thus worsen people's emotional well-being during difficult times. When people isolate themselves, they withdraw from the nourishment and support of others. Consequently, social contacts is considered as a habit which relies on an automatic and nonconscious enactment of the behavior, making it difficult to change [10].

Importantly, personal compliance with these two health behaviors strongly depends on several factors, such as social norms and pressure, attitudes towards the behaviors, perception of the ease to performs them and intentions. In this context, the present work aims at analyzing the application of handwashing and limitation of social contacts in the light of the theory of planned behavior.

## The Theory of Planned Behavior in the COVID-19 context

The Theory of Planned Behavior (TPB) posits that health behaviors can be predicted by intentions to perform them [11]. Intentions refer to the deliberate will to perform a behavior before applying it. These intentions, in turn, are affected a) by attitudes toward the behavior and whether people think it is useful, important or desirable, b) by the social norms they perceive to be prevailing around them, and c) by the control that people perceive they have over their actions. Thus, the TPB holds that attitudes, social norms and perceived control influence

behavior indirectly through intentions, such that intentions are the direct precursor of behavior. Furthermore, perceived behavioral control is hypothesized to affect behavior indirectly but also directly [12]. Similar to self-efficacy in the social cognitive theory proposed by Bandura [13], individuals with a strong sense of control approach difficult goals, have a strong commitment to their goals, maintain a task focus, persist in the face of failure, and attribute failure to a lack of effort [14]. Thus, a belief in one's ability to perform a behavior is likely to facilitate behavioral performance.

Several meta-analyses have provided evidence for the ability of the components of the TPB to predict general health behaviors (e.g., [15]). Interestingly, numerous studies have addressed these associations during the COVID-19 context, on several health behaviors such as physical distancing [16], mask wearing [17], handwashing [18], intentions of traveling [19], and intentions to receive a COVID-19 vaccine [20, 21]. In the same vein, Prasetyo and colleagues [22] demonstrated that factors derived from an extended version of TPB were significantly associated with behavioral intention (see also [23]). However, the literature suffers from a lack of studies focusing on limitation of social contacts as a health behavior. This limitation is addressed in the present research.

Specifically, we would expect that intentions, attitudes, social norms and the perceived control to apply handwashing will be greater than for limitation of social contacts, as the latter is characterized as an avoidant and unnatural behavior, that is more difficult to perform. Importantly, because the global situation varies significantly across countries, the present paper seeks to analyze the dynamics of these two health behaviors in two different national contexts, namely Belgium and France.

## COVID-19 in Belgium and France

Policies and strategies to face the COVID-19 pandemic show similar but also distinct characteristics in Belgium and in France. Figs 1 and 2 report the daily new confirmed COVID-19

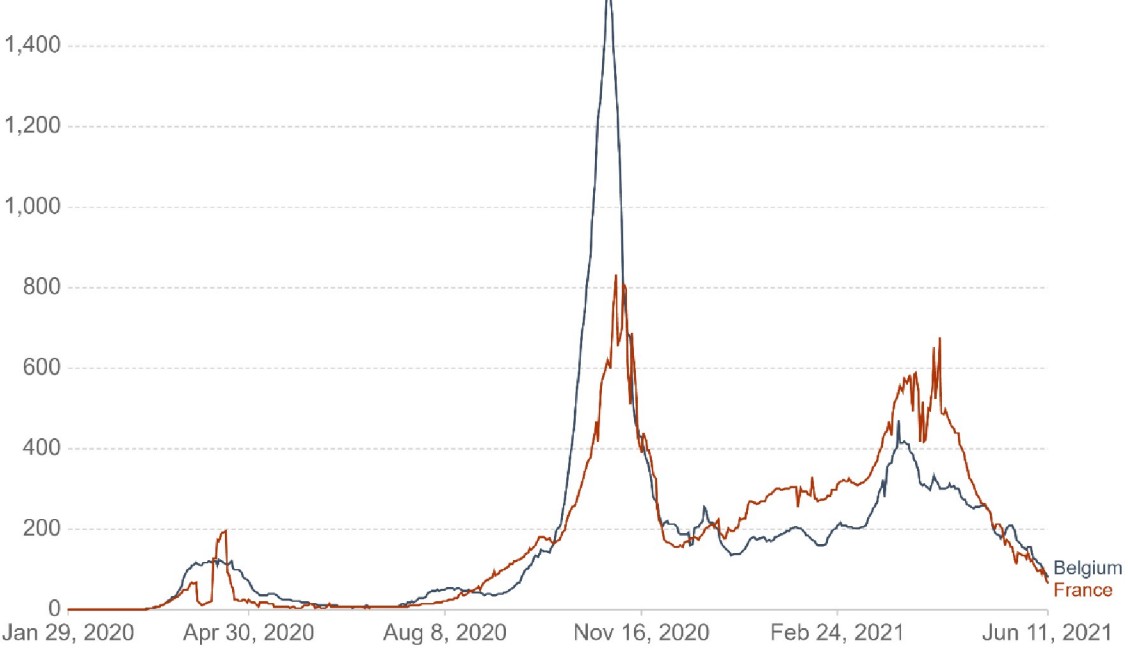

**Fig 1. Daily new confirmed COVID-19 cases per million people in Belgium and France.**

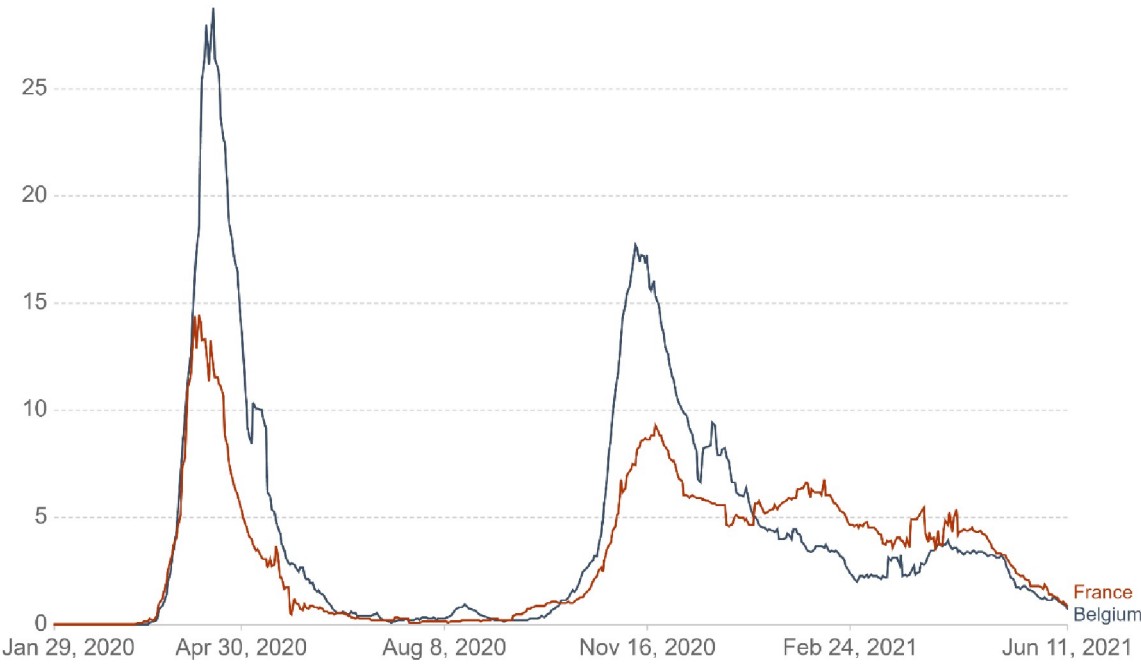

**Fig 2. Daily new confirmed COVID-19 deaths per million people in Belgium and France.**

cases and deaths per million people in both countries [24, 25]. As can be seen from these figures, Belgium and France experienced a peak in confirmed and death cases at the time of data collection (from March 18 until April 19, 2020). While both countries implemented several rules (e.g., closing schools and universities, non-food stores and close contact jobs) and adjusted them based on the national health situation, France's rules were more stringent at the time of data collection (see Fig 3). Specifically, France imposed stronger lockdowns with stricter curfews, more expensive fines for not complying with the regulation and longer travel restrictions. Notwithstanding their differences, Belgium and France actively follow public health recommendations (WHO, 2020b) that led them to improve their ability to cope with this challenging situation (i.e., flattening the curve). Despite these discrepancies between the two neighbors' countries, the high degree of compliance and the closeness regarding the national strategies adopted led us to expect that the TPB model should manifest similarly in Belgium and in France. Ultimately, while complying with health behaviors is crucial to handle new COVID-19 cases, a non-negligible proportion of both populations does not respect

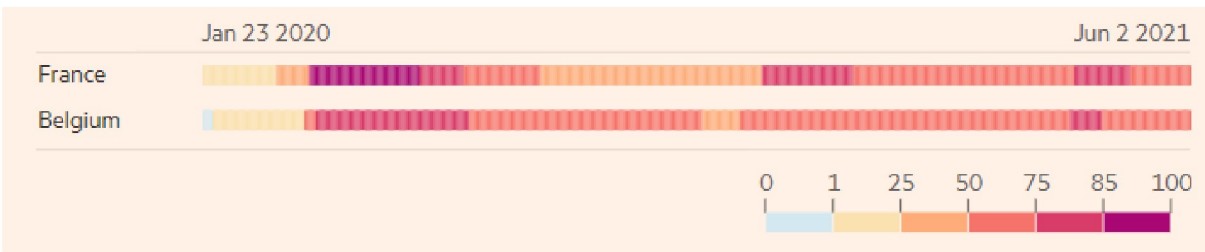

**Fig 3. COVID-19 government response stringency on lockdown in Belgium and France.**

handwashing and limitation of social contacts. Understanding the factors that play a key role in following sanitary behaviors is essential to better face global health threat such as the COVID-19 pandemic.

## Overview of the present work and hypotheses

The purpose of the present research is to compare the model of the TPB on handwashing and limitation of social contacts among people living in Belgium and France within the context of the COVID-19 pandemic.

**Hypothesis 1.** We would expect that intentions, attitudes, social norms and the perceived control to apply handwashing will be greater than for limitation of social contacts, as the latter is defined as an avoidant and unnatural behavior, that is more difficult to perform.

**Hypothesis 2.** We would expect to replicate the TPB model on both targeted health behaviors among both Belgian and French residents. First, we hypothesized that attitudes, social norms, and perceived control would positively predict intentions (Hypothesis 2a), which in turn would positively predict both handwashing and limitation of social contacts, independently in both Belgium and France (Hypothesis 2b). Next, we hypothesized that attitudes, social norms, and perceived control would indirectly predict both health behaviors through intentions (Hypothesis 2c). Finally, perceived control is hypothesized to also positively affect health behaviors directly, i.e., beyond and above the effect of intentions (Hypothesis 2d).

## Method

### Participants

A total of 4804 respondents participated in this study. Of them, 3744 indicated living in Belgium and 1060 in France. We decided to focus on the place of residence, instead of citizenship, because the place of living is more relevant here given the specific contexts of regulations that are organized within Belgium and France. Table 1 reports the sociodemographic characteristics for Belgian and French participants. Participants were recruited on the internet, through posts on social networks, advertisements on university pages and emails. They all completed the questionnaire voluntarily and without compensation. Data were collected from March 18 until April 19, 2020, which corresponds to the spring lockdown and the first peak of the pandemic in Belgium and France (for more information about the recruitment procedure, see [26]). This study has been approved by the ethics committee from the Research Institute for Psychological Sciences at Université catholique de Louvain (reference: Projet 2021–13). It has therefore been performed in accordance with the ethical standards laid down in the 1964 Declaration of Helsinki and its later amendments. Additionally, written informed consent was obtained from all participants.

**Table 1. Summary table of sociodemographic characteristics for Belgian and French participants.**

|  | Belgian | French | Total |
|---|---|---|---|
| *N* | 3744 | 1060 | 4804 |
| Age | 42.48 (16.69) | 37.09 (13.89) | 41.29 (16.27) |
| Gender | 74.6% women | 81.9% women | 76.2% women |
| Obtained a university diploma | 76.0% | 81.0% | 77.1% |
| Participant has been infected by COVID-19 | 9.8% | 12.5% | 10.4% |
| Family or friends have been infected by COVID-19 | 25.5% | 29.6% | 26.4% |
| Studying or practicing in a medical or paramedical field | 26.5% | 29.7% | 27.2% |

## Measures

The administered instruments were taken from previous studies and back translated to ensure construct equivalence [27]. The French version of the questionnaire, full data and executable RStudio and AMOS files including all the codes used to generate these analyses are available on the online supplementary material (https://osf.io/k3hfm/). Ultimately, participants had the additional option to report that the item did not apply to them. Consequently, we relied on listwise deletion to conduct analysis.

**Health behaviors.**   Two items assessed health behaviors. To measure handwashing, participants indicated how frequently they washed their hands on a Likert-scale (1 = *Never*; 2 = *1 to 5 times per day*; 3 = *6 to 10 times per day*; 4 = *11 to 15 times per day*; 5 = *More than 15 times a day*). To measure limitation of social contacts, participants indicated their level of agreement on a Likert-scale from 1 (*Completely disagree*) to 5 (*Completely agree)* with the statement "I limit my social contacts".

**Attitudes, social norms, perceived control, and intentions.**   To assess the central dimensions of the TPB [11], participants were asked to answer on a Likert-scale ranging from 1 (*Completely disagree*) to 5 (*Completely agree*) four items corresponding to the four dimensions for each of the two targeted health behaviors (handwashing and limitation of social contacts). Specifically, the four dimensions were attitudes ("I believe that doing *X* will limit spreading of the COVID-19" [*X* represents the targeted behavior]), social norms ("My relatives expect from me to do *X*"), perceived control ("For me, doing *X* is easy") and intentions ("I am ready to do *X*").

## Results

### Means differences and correlations

Tables 2 and 3 report the means, standard deviations and correlations for Belgian and French residents for handwashing and limitation of social contacts, respectively.

**Mean comparisons between health behaviors.**   To test Hypothesis 1, we conducted paired-sample *t*-tests to compare each TPB's dimensions (attitudes, social norms, perceived

**Table 2. Correlation matrix with Belgian above and French below, means and standard deviations for handwashing.**

|                    | Attitudes             | Social norms          | Perceived control     | Intentions            | Handwashing           |
|--------------------|-----------------------|-----------------------|-----------------------|-----------------------|-----------------------|
| Attitudes          | -                     | .35**                 | .31**                 | .39**                 | .21**                 |
| Social norms       | .31**                 | -                     | .29**                 | .31**                 | .16**                 |
| Perceived control  | .28**                 | .22**                 | -                     | .50**                 | .30**                 |
| Intentions         | .36**                 | .28**                 | .52**                 | -                     | .32**                 |
| Handwashing        | .17**                 | .13**                 | .28**                 | .29**                 | -                     |
| Belgium            | 4.40[a] (0.81)        | 3.98[a] (1.09)        | 4.39[a] (0.89)        | 4.66[a] (0.68)        | 3.08[a] (0.93)        |
| France             | 4.40[a] (0.83)        | 3.93[a] (1.09)        | 4.48[b] (0.86)        | 4.71[b] (0.62)        | 3.18[b] (0.95)        |
| Total              | 4.40 (0.81)           | 3.97 (1.09)           | 4.41 (0.88)           | 4.67 (0.66)           | 3.10 (0.93)           |
| Welch's *F*        | 0.00                  | 1.78                  | 9.01                  | 5.11                  | 7.58                  |
| *df*               | 1673.03               | 1701.87               | 1742.01               | 1845.34               | 1668.36               |
| p-value            | > .05                 | > .05                 | = .003                | = .024                | = .006                |
| Cohen's *d*        | .00                   | .05                   | .10                   | .08                   | .11                   |

Note:

**Correlation is significant at the 0.01 level. Standard deviations are presented in parentheses.

Values with the same superscript in the same column do not differ, *p* > 0.05.

**Table 3. Correlation matrix with Belgian above and French below, means and standard deviations for limitation of social contacts.**

|  | Attitudes | Social norms | Perceived control | Intentions | Social contacts |
|---|---|---|---|---|---|
| Attitudes | - | .47** | .14** | .50** | .23** |
| Social norms | .41** | - | .05** | .46** | .21** |
| Perceived control | .09** | .01 | - | .32** | .03 |
| Intentions | .42** | .38** | .35** | - | .22** |
| Social contacts | .25** | .17** | .03 | .21** | - |
| Belgium | 4.50ᵃ (0.83) | 4.04ᵃ (1.07) | 2.11ᵃ (1.32) | 3.94ᵃ (1.25) | 4.64ᵃ (0.79) |
| France | 4.58ᵇ (0.75) | 4.20ᵇ (0.98) | 2.36ᵇ (1.41) | 4.12ᵇ (1.14) | 4.65ᵃ (0.82) |
| Total | 4.52 (0.82) | 4.08 (1.05) | 2.17 (1.35) | 3.98 (1.23) | 4.64 (0.79) |
| Welch's $F$ | 10.05 | 21.52 | 27.65 | 21.24 | .07 |
| $df$ | 1857.80 | 1831.27 | 1628.01 | 1835.64 | 1643.45 |
| P-value | = .002 | < .001 | < .001 | < .001 | > .05 |
| Cohen's $d$ | .10 | .16 | .18 | .15 | .01 |

Note:

**Correlation is significant at the 0.01 level. Standard deviations are presented in parentheses.

Values with the same superscript in the same column do not differ, $p > 0.05$.

control and intentions) for the two health behaviors across groups. Results showed that intentions to apply the behavior ($t[3743] = 33.435$, $p < .001$, Cohen's $d = 0.55$ for Belgians and $t[1059] = 43.196$, $p < .001$, Cohen's $d = 0.50$ for French) and the perceived control over it ($t[3743] = 89.456$, $p < .001$, Cohen's $d = 1.46$ for Belgian and $t[1059] = 15.914$, $p < .001$, Cohen's $d = 1.33$ for French) were significantly and largely higher for handwashing than for limiting social contacts, across both groups. In contrast, participants had a more positive attitudes ($t[3743] = -6.051$, $p < .001$, Cohen's $d = 0.10$ for Belgian and $t[1059] = -6.216$, $p < .001$, Cohen's $d = 0.19$ for French) and felt more pressure from social norms ($t[3743] = -2.838$, $p = .005$, Cohen's $d = 0.05$ for Belgian and $t[1059] = -7.504$, $p < .001$, Cohen's $d = 0.23$ for French) regarding their limitation of social contacts than their handwashing, in both groups.

Importantly, to assess the importance of the mean differences described above, we calculated Cohen's $d$ and based our interpretations on Cohen's thresholds with values around 0.2 indicating small effects, 0.5 for medium effects, and around 0.8 for large effects [28]. Consequently, it is pertinent to note that differences on intentions and perceived control were large (greater for handwashing), whereas the effect of attitudes and social norms were small (greater for limitation of social contacts).

**Mean comparisons between countries of residence.** Following the recommendations on statistical practice [29], Welch's analysis of variance was conducted to compare the different variables for Belgian and French residents. First, we found a main effect of residence on the targeted health behavior of handwashing, with French indicated washing their hands more frequently than Belgian residents. Second, we observed main effects of residence on perceived control and intentions, indicating that French were more willing to wash their hands and consider this behavior easier to do, as compared to Belgian residents. Ultimately, there were no difference for attitudes and social norms between groups (see Table 2).

When looking at the second targeted health behavior, we found that French and Belgian residents reported limiting their social contacts in a similar way ($p > .05$). Interestingly, we also observed that French residents were reporting higher scores of attitudes, social norms, perceived control, and intentions, related to the limitation of social contacts, than Belgian

residents (see Table 3). Importantly, it is relevant to note that all significant effects between Belgium and France described above were small (see Tables 2 and 3).

**Correlations.** As expected, we observed that all dimensions of the TPB are positively intercorrelated, and positively associated with the targeted health behavior of handwashing, for both Belgian and French residents (see Table 2). Turning now to limitation of social contacts, the same pattern of correlations emerged, except that perceived control was not associated with the second targeted health behavior for both groups. Interestingly, perceived control was neither associated with social norms for French residents (see Table 3).

## Multi-group modeling

**Analysis strategy.** To test the paths developed in the TPB model among Belgian and French residents, we conducted a path analysis testing Hypothesis 2 regarding direct and indirect relations on both targeted health behaviors, independently. Attitudes, social norms, and perceived behavior were the predictors, intentions was the mediator and handwashing and limitation of social contacts were the outcome variables. The model fit was assessed using the cut-off criteria of $CFI > .90$ and $RMSEA < .08$ [30].

Following the recommendations for analyzing the TPB model using path analysis [31], we first analyzed a saturated model including all possible paths in the model to calculate direct effects of each predictor. Then, based on theory, we deleted the two direct paths from attitudes to behavior and social norms to behavior to avoid saturated model, retaining only the expected direct effect of perceived control on health behavior. Specifically, we removed the direct paths from attitudes to limitation of social contacts (Belgium: $B = .126$, $SE = .018$, $p < .001$; France: $B = .179$, $SE = .037$, $p < .001$), social norms to limitation of social contacts (Belgium: $B = .099$, $SE = .014$, $p < .001$; France: $B = .055$, $SE = .028$, $p > .05$), attitudes to handwashing (Belgium: $B = .073$, $SE = .020$, $p < .001$; France: $B = .060$, $SE = .037$, $p > .05$) and social norms to limitation of social contacts (Belgium: $B = .023$, $SE = .014$, $p > .05$; France: $B = .021$, $SE = .027$, $p > .05$). We thus re-estimated the model with these two paths dropped and reported the resulting coefficients below.

Additionally, we tested the equality of correlations and standardized regression coefficients between Belgian and French residents using a series of Wald Chi-Squared Tests [32]. Specifically, we assessed constraints on statistical parameters (i.e., correlations and regressions paths individually and independently) based on the weighted distance between the unrestricted estimate and its hypothesized value under the null hypothesis, where the weight is the precision of the estimate [33, 34].

**Handwashing.** When considering handwashing as the dependent variable, the multi-group model provided excellent model fit statistics ($\chi2 = 23.995$, $df = 4$, $CFI = .995$, $RMSEA = .032$). Standardized path coefficients for Belgian and French residents are shown in Fig 4. In line with the TPB, the three dimensions of the TPB were strongly and positively associated with each other for both Belgian and French participants. Importantly, Wald Chi-Squared Tests revealed that the correlation between social norms and perceived control was significantly stronger among Belgian people ($\Delta\chi^2 = 4.825$, $p = .028$). Furthermore, and in line with Hypothesis 2a, we observed that the three dimensions of TPB positively predicted intentions. Note that the path from attitudes to intentions was significantly stronger among Belgian people ($\Delta\chi^2 = 3.975$, $p = .046$). Next, and as expected (Hypothesis 2b), we found that intentions positively and similarly predicted handwashing for both groups.

To complement our statistical analysis, we explored the indirect effects of attitudes, social norms, and perceived control on handwashing through intentions within both groups. In line with Hypothesis 2c, all indirect effects were significant. Specifically, the indirect effect of

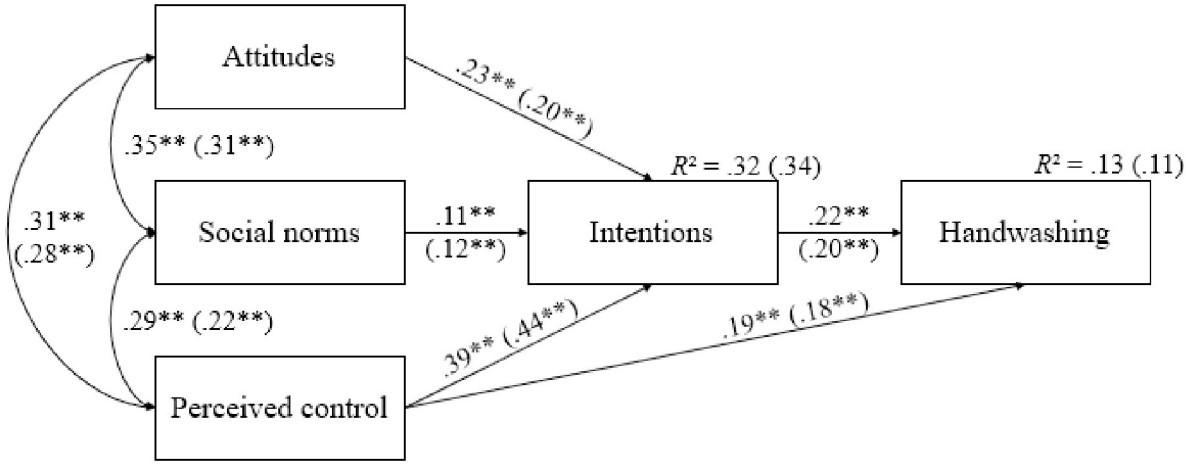

**Fig 4. Standardized path coefficients for Belgian and French (in parentheses) residents for handwashing ($N$ = 3744).** $^{**}$ $p$ < .001. $R^2$ = R-squared.

perceived control on handwashing through intentions (Belgium: $B$ = .088, $SE$ = .008, $p$ < .001; France: $B$ = .088, $SE$ = .018, $p$ < .001) was significant for both groups, followed by attitudes (Belgium: $B$ = .052, $SE$ = .006, $p$ < .001; France: $B$ = .040, $SE$ = .010, $p$ < .001) and then social norms (Belgium: $B$ = .025, $SE$ = .003, $p$ < .001; France: $B$ = .024, $SE$ = .006, $p$ < .001). Ultimately, and in line with Hypothesis 2d, the direct effect of perceived control on handwashing remained significant for both Belgian and French residents when controlling for intentions (see Fig 4).

**Limitation of social contacts.** When considering limitation of social contacts as the dependent variable, the multigroup model provided good model fit statistics ($\chi2$ = 153.913, $df$ = 4, $CFI$ = .960, $RMSEA$ = .079). Standardized path coefficients for Belgian and French residents are shown in Fig 5. Consistent with the TPB, the three dimensions of the TPB were strongly and positively associated with each other for both Belgian and French residents, except the correlation between perceived control and social norms that did not yield significance in the French sample. Note however that the strength of this correlation did not differ

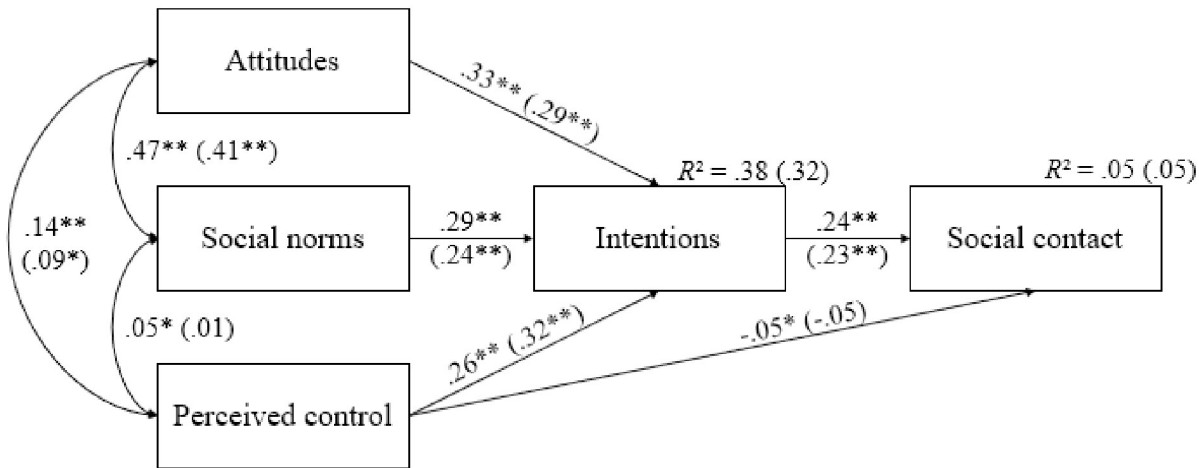

**Fig 5. Standardized path coefficients for Belgian and French (in parentheses) residents for limitation of social contacts ($N$ = 3744).** $^{*}$ $p$ < .01. $^{**}$ $p$ < .001. $R^2$ = R-squared.

between groups ($\Delta\chi^2$ = 1.391, $p > .05$). Importantly, Wald Chi-Squared Tests revealed that the correlation between attitudes and social norms ($\Delta\chi^2$ = 14.103, $p < .001$) was significantly stronger among Belgian residents. Furthermore, and in line with Hypothesis 2a and Hypothesis 2b, we observed that the three dimensions of TPB positively predicted intentions, which in turn positively predicted social contacts for both Belgian and French residents, similarly.

To complement our statistical analysis, we explored the indirect effects of attitudes, social norms, and perceived control on limitation of social contacts through intentions within both groups. In line with Hypothesis 2c, the indirect effect of attitudes (Belgium: $B = .077$, $SE = .006$, $p < .001$; France: $B = .067$, $SE = .012$, $p < .001$) followed by social norms (Belgium: $B = .068$, $SE = .004$, $p < .001$; France: $B = .056$, $SE = .008$, $p < .001$), then perceived control (Belgium: $B = .060$, $SE = .003$, $p < .001$; France: $B = .073$, $SE = .007$, $p < .001$), were all significant. Ultimately, and contrary to Hypothesis 2d, we found that perceived control directly and negatively predicted limitation of social contacts in Belgium, whereas this path was not significant in France. However, this effect was similar for both national groups ($\Delta\chi^2$ = .945, $p > .05$) and extremely small, as compared to all the others coefficient paths (see Fig 5).

Ultimately, it is pertinent to note that the present model has been replicated on both targeted health behaviors among Belgian and French residents, while controlling for age, gender, education, being infected by COVID-19 and relatives being infected.

## Discussion

Although the vast majority of studies have focused on the detrimental physical and psychological consequences of the COVID-19 pandemic, very little is known regarding the mechanisms that improve the adoption and maintenance of health behaviors, which are crucial for slowing the spread of infectious diseases. Relying on a cross-national comparison research design, the current work is the first to analyze the TPB model on handwashing and limitation of social contacts among people living in Belgium and France within the context of the COVID-19 pandemic. Importantly, the results of the present study indicate that (1) the four TPB components play a key role in developing and maintaining health behaviors during the COVID-19 crisis and (2) suggest that the TPB model tends to manifest similarly across countries in explaining protective behaviors, when comparing different targeted health behaviors among individuals living in different national contexts.

### TPB model in Belgium and France

When faced with a pandemic, it is essential to highlight which factors can lead to the adherence to health behaviors. In this context, the TPB has been widely used to understand the adoption of protective and sanitary behaviors in response to health problems [35, 36]. In this study, we used multi-group modeling to examine how the TPB's dimensions were related to handwashing and limitation of social contacts in the context of the COVID-19 sanitary crisis. Taking minor variations into account, we successfully replicated the TPB model on handwashing and limitation of social contacts among both Belgian and French residents. In sum, we found that more positive attitudes, greater social pressure, increased perceived control and higher intentions were related to higher adherence to the two health behaviors for both national groups. This is consistent with a flourishing number of studies demonstrating their key role to foster preventive and protective behaviors in the context of the COVID-19 pandemic (e.g., [21, 37]). First, we observed that each person's favorable or unfavorable evaluation of a health behavior shapes outcomes associated with it (i.e., attitudes). Second, we found strong influence of the way each individual's beliefs about whether their reference groups approve or disapprove of the health behavior and whether they are motivated to comply with their reference group's

norms (i.e., social norms). Third, we evidenced that favorable attitudes, combined with the approval of the behavior by others close to them, strengthens the intentions to perform that particular health behavior (i.e., intentions), which in turn positively influenced both handwashing and limitation of social contacts. The results of the present study are consistent with other studies who also demonstrated that positive attitudes (e.g., [20]), social norms (e.g., [38, 39]), and intentions (e.g., [40, 41] play key roles in the adhesion of health behaviors related to the COVID-19 (see also [22, 42]).

Interestingly, we observed that the person's perceived ability to perform a health behavior (i.e., perceived control) also indirectly predicted both targeted health behaviors through intentions for both Belgian and French residents. However, even if this belief increases intentions to perform the two health behaviors, perceived control was only *directly* associated with the actual behavioral application (self-reported) of handwashing and not with limitation of social contacts. Specifically, the coefficient path from perceived control to limitation of social contacts, while controlling for intentions, was significant and negative for Belgians, whereas it was not significant for French residents. However, the size of this effect was extremely small and neglectable. More precisely, while the indirect effects remained significant, perceived control played a limited role on limitation of social contacts, as compared to the other components of the TPB. This is consistent with our results indicating that the perceived control was significantly and largely lower for limiting social contacts than for handwashing, across both groups, even though participants had more positive attitudes and felt more social pressure regarding their limitation of social contacts, as compared to handwashing. These findings came as no surprise because limitation of social contacts is an unnatural and counterintuitive behavior which makes it more difficult to apply as it pushes against people's instinctual need for connectedness [9], whereas handwashing is much easier to apply and is already considered as a habit [7, 10].

Furthermore, Belgian and French residents demonstrated similar patterns of responses, with minor differences that only and weakly differed in terms of magnitude and not in direction, suggesting that the TPB model manifested similarly across both residential groups. Importantly, these results have been replicated by controlling for age, gender, education, being infected by COVID-19 and relatives being infected, which confirms the robustness of our findings. While other researchers have also found little variations from the original TPB model (e.g., [43]), the vast majority of studies concluded that the TPB provides a valid theory for understanding the determinants of health behavior and developing behavioral interventions [12]). In sum, the findings of the present study confirm and support this assumption.

## Implications and future directions

Research on COVID-19 is flourishing. The associations found in the present study enrich the current literature and suggest implications for research and practice.

**Research.**   The literature exploring the role of TPB during the COVID-19 pandemic is flourishing. However, there is a lack of research focusing on limitation of social contacts as a health behavior. This study overcomes this limitation and affords a robust validation of the TPB as a conceptual framework for explaining the adoption of handwashing and limitation of social contacts. The influences from the TPB dimensions in COVID-19 context advocate attitudes, social norms, perceived control and intentions as viable determinants of such health behaviors [44, 45]. Although that the current study reinforces TPB, the percentage of variance explained for both targeted behaviors remained low (i.e., around 10%), suggesting that potential additional dimensions such as cooperation [46], emotions [25], resilience [47], and self-compassion [48] should be considered in order to improve the behavior prediction.

**Practice.**   Even though more people are getting the COVID-19 vaccine, pandemic precautions like handwashing, limitation of social contacts, physical distancing and wearing masks are still essential to slow the spread of the virus. While vaccines are a crucial tool in beating the virus, they cannot be used alone at this time. Because a favorable evaluation of health behaviors leads to their application within the population, governments, policymakers, health institutions, and experts should find out communication plans that support increasing willingness of people through favorable attitudinal changes toward such health behaviors. These communication strategies should encourage COVID-19 preventive behaviors while taking into account the role of family, peers and colleagues in developing and maintaining health behaviors, as well as the difficulty for some individuals to apply some of these behaviors (e.g., inhibition of natural tendencies to interact with others).

**Limitations.**   Although this study expands our understanding of how the TPB components are associated with health behaviors in the COVID-19 context, this study is not without limitations. First, although the original TPB questionnaire contains several items for each component, we relied on a 4-item version with one item per dimension reused from other studies [25]. As in many COVID-19 related studies, reducing scales is a common practice that allows researchers to explore a wide variety of constructs while avoiding participants' fatigue. However, these reduced scales might not capture the type of attitudes, social norms, perceived control and intentions that triggers health behavior responses during the pandemic. For instance, our measure of attitudes (i.e., I believe that doing handwashing/limitation of social contacts will limit spreading of the COVID-19) only assessed the facet of outcome efficacy, which refers to the degree to which a person evaluates in a favorable or unfavorable way the behavior in question, and whether the implemented behavior will lead to positive or negative outcomes [49]. Similarly, the two targeted behaviors were also measured using a single item for each, which makes it difficult to adequately assess all types and processes of such behaviors among individuals. Future work should explore all facets of these constructs, to accurately capture manifestations of health behaviors as they may vary among people.

Second, the sample sizes vary between Belgium and France. This has two important implications: the estimations of the relations between variables are largely determined by samples with greater sample sizes (i.e., Belgium, $N = 3744$) than those with smaller sample sizes (i.e., France, $N = 1060$). Specifically, as the sample size increases, the confidence in the estimate and model fit statistics increases, and the uncertainty decreases which lead to greater precision. In the same vein, the estimation of the true value of the mean in the population is more precise in the more heavily represented country (i.e., Belgium), but it is relevant to note that a determination of sample size by statistical power analysis demonstrated that a sample size of 160 per group is generally deemed excellent. Because this criterion is too often neglected in the current literature, we encourage scholars to systematically rely on large and close sample sizes in the context of cross-national research to more accurately capture national differences.

Third, the current study compares the original TPB model in Belgium and France. However, studies have explored extended TPB models including additional components (e.g., perceived vulnerability, perceived severity, knowledge about COVID-19, risk perception of COVID-19). These extended components could play a role in explaining behavioral responses in the French and Belgian national contexts [22, 23].

## Conclusion

The current pandemic has affected everyone and changed the ways in which each of us relates to and navigates the world. Even though the arrival of COVID-19 vaccines marked a turning point in the pandemic, health behaviors are still necessary to prevent the spread of the virus.

Because self-imposed preventive measures can drastically impact the rate of infection among populations, adopting and maintaining health behaviors is essential to cope with the COVID-19 pandemic. While specific communication strategies should be encouraged to promote preventive behaviors, they should consider the role of attitudes, social pressure and perceived ability in order to find a balance between physical and mental health.

## Author Contributions

**Conceptualization:** Robin Wollast, Mathias Schmitz, Alix Bigot, Olivier Luminet.

**Data curation:** Mathias Schmitz.

**Formal analysis:** Robin Wollast.

**Funding acquisition:** Olivier Luminet.

**Methodology:** Robin Wollast, Alix Bigot, Olivier Luminet.

**Project administration:** Olivier Luminet.

**Supervision:** Olivier Luminet.

**Writing – original draft:** Robin Wollast.

**Writing – review & editing:** Robin Wollast, Mathias Schmitz, Alix Bigot, Olivier Luminet.

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
