## [Decision Letter · Decision Letter 0]

19 Aug 2021

PONE-D-21-20498

The theory of planned behavior during the COVID-19 pandemic:

A comparison of health behaviors between Belgian and French residents

PLOS ONE

Dear Dr. Wollast,

Thank you for submitting your manuscript to PLOS ONE. After careful consideration, we feel that it has merit but does not fully meet PLOS ONE’s publication criteria as it currently stands. Therefore, we invite you to submit a revised version of the manuscript that addresses the points raised during the review process.

Please find below the reviewers' comments, as well as those of mine.

We look forward to receiving your revised manuscript.

Kind regards,

Valerio Capraro

Academic Editor

PLOS ONE

Journal Requirements:

"This study has been supported by a grant from the Foundation Louvain."

"This study has been supported by a grant from the Foundation Louvain."

"This study has been supported by a grant from the Foundation Louvain" 

Additional Editor Comments (if provided):

I have now collected two reviews from two experts in the field. Both reviewers think that the paper has potential, but suggest several improvements. Therefore, I would like to invite you to revise your work for Plos One. Needless to say that all comments should be addressed. Besides the reviewers' comments, I would like to add a couple more comments. The relationship between social norms and protective behaviour was investigated also in other papers (e.g., Bilancini et al. 2020). The "perspective article" on what behavioural science can do to support pandemic response, published by Van Bavel et al. in Nature Human Behaviour, might be a useful general reference. Of course, it is not a requirement to cite these papers, but in general I think that the paper should be placed better within the emerging literature.

I am looking forward for the revision.

Bilancini E, Boncinelli L, Capraro V, Celadin T, Di Paolo R (2020) The effect of norm-based messages on reading and understanding COVID-19 pandemic response governmental rules. Journal of Behavioral Economics for Policy 4, Special Issue 1, 45-55.

Van Bavel, J. J., et al. (2020). Using social and behavioural science to support COVID-19 pandemic response. Nature Human Behaviour, 4, 460-471.

Reviewers' comments:

Reviewer's Responses to Questions

**Comments to the Author**

1. Is the manuscript technically sound, and do the data support the conclusions?

Reviewer #1: Yes

Reviewer #2: Yes

2. Has the statistical analysis been performed appropriately and rigorously? 

Reviewer #1: No

Reviewer #2: Yes

3. Have the authors made all data underlying the findings in their manuscript fully available?

Reviewer #1: Yes

Reviewer #2: Yes

4. Is the manuscript presented in an intelligible fashion and written in standard English?

Reviewer #1: Yes

Reviewer #2: Yes

5. Review Comments to the Author

Reviewer #1: The study examines The present study affords a validation of the Theory of Planned Behavior (TPB) as a conceptual framework for explaining the adoption of handwashing and limitation of social contacts, two health behaviors that highly differ in their nature. I believe this manuscript has substantial contributions in the public health field, however, there are several questions and doubts the authors need to address in a major revision. Here are some:

1. The proposed conceptual framework lacks in-depth discussion. How did the authors come up with the latent constructs in their model? What makes their model different from similar studies and how did they ensure the validity of the content? There was one comprehensive study by Prasetyo et al., (2020) who also explored factors affecting the perceived effectiveness of COVID-19 prevention measures by utilizing PMT and extended TPB. This manuscript needs to discuss in-depth the differences between Prasetyo et al., (2020) in the introduction and discussion parts.

• Prasetyo, Y., Castillo, A., Salonga, L., Sia, J., & Seneta, J. (2020). Factors affecting perceived effectiveness of COVID-19 prevention measures among Filipinos during Enhanced Community Quarantine in Luzon, Philippines: Integrating Protection Motivation Theory and extended Theory of Planned Behavior. International Journal of Infectious Diseases, 99, 312-323. https://nam05.safelinks.protection.outlook.com/?url=https%3A%2F%2Fdoi.org%2F10.1016%2Fj.ijid.2020.07.074&data=04%7C01%7Cytprasetyo%40mapua.edu.ph%7C2781c2ff714741f6f3ba08d88bdd4cb0%7Cc7e8b5ac96c64123a65a793543aced4d%7C0%7C0%7C637413129903670551%7CUnknown%7CTWFpbGZsb3d8eyJWIjoiMC4wLjAwMDAiLCJQIjoiV2luMzIiLCJBTiI6Ik1haWwiLCJXVCI6Mn0%3D%7C1000&sdata=L2UPUGnmfP9OhWjeIx39tkUHzweCdcJX3%2BmL7GJmYqo%3D&reserved=0

2. Similarly, the research gap of this study in is lacking in most updated studies. I cannot justify in what way(s) the past researches failed to provide valuable information regarding this matter. Please add some of the most updated studies related to TPB in the context of COVID-19.

3. Please show the mean, standard deviation, and the factor loading for each indicator.

4. Please show the demographic characteristic of the respondents by using a table, not just the description.

5. Please also show each indicator under the latent in one table, not just the description. It will enhance the readability of the paper.

6. I want to see the reliability and validity test results. Please show the cronbach's alpha and composite reliability.

7. Why splitting handwashing and social contacts? Why not combine it in 1 model with 2 exogenous latent variables?

8. Why not adding the country in the SEM? Perhaps you can do the dummy code 1: French 2: Belgium. Let’s see the effect of the country in the model. Or you may split the model into 2: Fench model and Belgium model.

9. Again, there are many TPB studies in 2021. Please compare your findings with the most updated studies.

Reviewer #2: Congratulations on this well written piece of work. Please find my comments below;

"Several meta-analyses have provided evidence for the ability of the components of the

TPB to predict general health behaviors (e.g., Conner & Norman, 2015). Interestingly, a few

studies have addressed these associations during the COVID-19 context, on several health

behaviors such as physical distancing (Zhang et al., 2019), mask wearing (Chung et al., 2018)

and intentions to receive a COVID-19 vaccine (Chu & Liu, 2021; Shmueli, 2021). However,

the literature suffers from a lack of studies focusing on limitation of social contacts as a health

behavior. This limitation is addressed in the present research."

- While that is true to an extent, you may want to have a look at this article published in PLOS1 that has assessed the behavior modifications during COVID-19 using the TPB on a wide multi-country scale (https://journals.plos.org/plosone/article?id=10.1371/journal.pone.0239961)

"suggest that the TPB model is a more universal rather than culture specific process in explaining protective behaviors, when comparing different targeted health behaviors among individuals living in different national contexts."

- I believe that may be an overstatement. Verily, Belgian and French nationals do differ appreciably with respect to culture and social norms, however, that difference is not striking, given their very close geographic location, european heritage and descent.

- I am under the impression that one of the aims of this manuscript was to attest to the validity of the TPB, which has been extensively used and validated in the fiield of health research since its advent in 1991.

6. PLOS authors have the option to publish the peer review history of their article (what does this mean?). If published, this will include your full peer review and any attached files.

Reviewer #1: **Yes: **Assoc.Prof.Yogi Tri Prasetyo, Ph.D.

Reviewer #2: No

---

## [Author Response · Author response to Decision Letter 0]

7 Sep 2021

Reviewers' comments:

Reviewer's Responses to Questions

Comments to the Author

1. Is the manuscript technically sound, and do the data support the conclusions?

Reviewer #1: Yes

Reviewer #2: Yes

2. Has the statistical analysis been performed appropriately and rigorously?

Reviewer #1: No

Reviewer #2: Yes

3. Have the authors made all data underlying the findings in their manuscript fully available?

Reviewer #1: Yes

Reviewer #2: Yes

4. Is the manuscript presented in an intelligible fashion and written in standard English?

Reviewer #1: Yes

Reviewer #2: Yes 

5. Review Comments to the Author

Reviewer #1: The study examines The present study affords a validation of the Theory of Planned Behavior (TPB) as a conceptual framework for explaining the adoption of handwashing and limitation of social contacts, two health behaviors that highly differ in their nature. I believe this manuscript has substantial contributions in the public health field, however, there are several questions and doubts the authors need to address in a major revision. Here are some:

Response: We would like to thank Associate Professor Yogi Tri Prasetyo for his careful read on our manuscript and constructive comments.

1. The proposed conceptual framework lacks in-depth discussion. How did the authors come up with the latent constructs in their model? What makes their model different from similar studies and how did they ensure the validity of the content? There was one comprehensive study by Prasetyo et al., (2020) who also explored factors affecting the perceived effectiveness of COVID-19 prevention measures by utilizing PMT and extended TPB. This manuscript needs to discuss in-depth the differences between Prasetyo et al., (2020) in the introduction and discussion parts.

• Prasetyo, Y., Castillo, A., Salonga, L., Sia, J., & Seneta, J. (2020). Factors affecting perceived effectiveness of COVID-19 prevention measures among Filipinos during Enhanced Community Quarantine in Luzon, Philippines: Integrating Protection Motivation Theory and extended Theory of Planned Behavior. International Journal of Infectious Diseases, 99, 312-323. https://www.ijidonline.com/article/S1201-9712(20)30622-6/fulltext 

Response: Thank you for the appreciation of our work. The purpose of the present research was to compare the model of the TPB on handwashing and limitation of social contacts among people living in Belgium and France within the context of the COVID-19 pandemic. To assess the central dimensions of the TPB (Ajzen, 1991), participants were asked to answer on a Likert-scale ranging from 1 (Completely disagree) to 5 (Completely agree) four items corresponding to the four dimensions for each of the two targeted health behaviors (handwashing and limitation of social contacts). Specifically, the four dimensions were attitudes (“I believe that doing X will limit spreading of the COVID-19” [X represents the targeted behavior]), social norms (“My relatives expect from me to do X”), perceived control (“For me, doing X is easy”) and intentions (“I am ready to do X”). Consequently, it is important to highlight that we only used one item per dimension, which was adapted for each health behavior. This design makes it statistically impossible to rely on a model with latent constructs including factor loadings. This is why we reported a path analysis of observed variables. However, we validated our model in both Belgium and France using multiple fit indices (e.g., CFI, RMSEA) to provide a multifaceted assessment of the models (Tanaka, 1993). Model fit statistics were good suggesting that our model holds in both national contexts. 

As suggested, we have included the work of Prasetyo et al. (2020) in our research. Specifically, we wrote that: “Interestingly, numerous studies have addressed these associations during the COVID-19 context, on several health behaviors such as physical distancing (Zhang et al., 2019), mask wearing (Kim et al., 2020), handwashing (Ammar et al., 2020), intentions of traveling (Hamid & Bano, 2021), and intentions to receive a COVID-19 vaccine (Chu & Liu, 2021; Shmueli, 2021). In the same vein, Prasetyo et al. (2020) demonstrated that factors derived from an extended version of TPB were significantly associated with behavioral intention (see also Fan et al., 2021).”

Additionally, we have also referred to their work in the discussion: “The results of the present study are consistent with other studies who also demonstrated that positive attitudes (e.g., Chu & Liu, 2021), social norms (e.g., Capraro & Barcelo, 2020; Lunn et al., 2020), and intentions (e.g., Gibson et al., 2021; Zhang et al., 2020) play key roles in the adhesion of health behaviors related to the COVID-19 (see also Koi & Long, 2021; Prasetyo et al., 2020).”

Finally, we wrote that: “Third, the current study compares the original TPB model in Belgium and France. However, studies have explored extended TPB models including additional components (e.g., perceived vulnerability, perceived severity, knowledge about COVID-19, risk perception of COVID-19). These extended components could play a role in explaining behavioral responses in the French and Belgian national contexts (Fan et al., 2021; Prasetyo et al., 2020).”

2. Similarly, the research gap of this study in is lacking in most updated studies. I cannot justify in what way(s) the past researches failed to provide valuable information regarding this matter. Please add some of the most updated studies related to TPB in the context of COVID-19.

Response: As suggested, we have included several studies related to TPB, or specific components of the TBP (social norms, attitudes, perceived control, intentions) in the context of COVID-19. Specifically: 

Ammar, N., Aly, N. M., Folayan, M. O., Khader, Y., Virtanen, J. I., Al-Batayneh, O. B., … El Tantawi, M. (2020). Behavior change due to COVID-19 among dental academics—The theory of planned behavior: Stresses, worries, training, and pandemic severity. PLOS ONE, 15(9), e0239961. https://doi.org/10.1371/journal.pone.0239961

Capraro, V., & Barcelo, H. (2020, May 16). The effect of messaging and gender on intentions to wear a face covering to slow down COVID-19 transmission. PsyArXiv. Retrieved from https://doi.org/10.31234/osf.io/tg7vz

Fan, C.-W., Chen, I.-H., Ko, N.-Y., Yen, C.-F., Lin, C.-Y., Griffiths, M. D., & Pakpour, A. H. (2021). Extended theory of planned behavior in explaining the intention to COVID-19 vaccination uptake among mainland Chinese university students: an online survey study. Human Vaccines & Immunotherapeutics, 1–8. https://doi.org/10.1080/21645515.2021.1933687

Hossain, M. B., Alam, M. Z., Islam, M. S., Sultan, S., Faysal, M. M., Rima, S., … Al Mamun, A. (2021). Health Belief, Planned Behavior, or Psychological Antecedents: What predicts COVID-19 Vaccine Hesitancy better among the Bangladeshi Adults? https://doi.org/10.1101/2021.04.19.21255578

Kim, Y.-J., Cho, J., & Kang, S.-W. (2020). Study on the Relationship between Leisure Activity Participation and Wearing a Mask among Koreans during COVID-19 Crisis: Using TPB Model. International Journal of Environmental Research and Public Health, 17(20), 7674. https://doi.org/10.3390/ijerph17207674

Khoi, B. H., & Long, N. N. (2020). An Empirical Study about the Intention to Hoard Food during COVID-19 Pandemic. Eurasia Journal of Mathematics, Science and Technology Education, 16(7), em1857. doi:10.29333/ejmste/8207

Lunn, P. D., Timmons, S., Belton, C. A., Barjaková, M., Julienne, H., & Lavin, C. (2020). Motivating social distancing during the COVID-19 pandemic: An online experiment. Social Science & Medicine, 265, 113478. https://doi.org/10.1016/j.socscimed.2020.113478

Gibson, L. P., Magnan, R. E., Kramer, E. B., & Bryan, A. D. (2021). Theory of Planned Behavior Analysis of Social Distancing During the COVID-19 Pandemic: Focusing on the Intention–Behavior Gap. Annals of Behavioral Medicine, 55(8), 805–812. https://doi.org/10.1093/abm/kaab041

Hamid, S., & Bano, N. (2021). Behavioral Intention of Traveling in the period of COVID-19: An application of the Theory of Planned Behavior (TPB) and Perceived Risk. International Journal of Tourism Cities. https://doi.org/10.1108/ijtc-09-2020-0183

Prasetyo, Y., Castillo, A., Salonga, L., Sia, J., & Seneta, J. (2020). Factors affecting perceived effectiveness of COVID-19 prevention measures among Filipinos during Enhanced Community Quarantine in Luzon, Philippines: Integrating Protection Motivation Theory and extended Theory of Planned Behavior. International Journal of Infectious Diseases, 99, 312-323. https://www.ijidonline.com/article/S1201-9712(20)30622-6/fulltext 

Zhang, M., Li, Q., Du, X., Zuo, D., Ding, Y., Tan, X., & Liu, Q. (2020). Health behavior toward COVID-19: the role of demographic factors, knowledge, and attitude among Chinese college students during the quarantine period. Asia Pacific Journal of Public Health, 32(8), 533-535. https://doi.org/10.1177/1010539520951408

3. Please show the mean, standard deviation, and the factor loading for each indicator.

Response: Means, standard deviations and related statistics for both Belgian and French participants are reported in Table 2 for handwashing and Table 3 for limitation of social contacts. Factor loadings are not included in our model as we do not rely on a model with latent constructs (see Response 1). 

4. Please show the demographic characteristic of the respondents by using a table, not just the description.

Response: As suggested, we have created a new table including demographic characteristics for both Belgian and French participants (see Table 1).

5. Please also show each indicator under the latent in one table, not just the description. It will enhance the readability of the paper.

Response: We did not rely on a latent analysis (see Response 1). 

6. I want to see the reliability and validity test results. Please show the cronbach's alpha and composite reliability.

Response: Given that we have one item per dimension, we cannot calculate Cronbach's alphas. However, an inspection of the correlations revealed strong, significant and positive associations between the main components of the TPB among both national groups and for both health behaviors, indicating high reliabilities among the selected items.

7. Why splitting handwashing and social contacts? Why not combine it in 1 model with 2 exogenous latent variables?

Response: As described in the method section (see also Response 1), we relied on different items for handwashing and limitation of social contacts, which does not allow to include them in the same model. Most importantly, adapting our items to the specific measures of health behaviors allowed us to better capture their associations with handwashing and limitation of social contacts.

8. Why not adding the country in the SEM? Perhaps you can do the dummy code 1: French 2: Belgium. Let’s see the effect of the country in the model. Or you may split the model into 2: Fench model and Belgium model.

Response: This is a good suggestion. We have indeed relied on your second suggestion and tested the model as a function of nationality. We selected this statistical method because it allowed us to test the equality of correlations and standardized regression coefficients between Belgian and French residents using a series of Wald Chi-Squared Tests (Klopp, 2019). We believe it is the most robust, modern and appropriate statistical approach to adequately capture the strength of our data. 

9. Again, there are many TPB studies in 2021. Please compare your findings with the most updated studies.

Response: We agree with you that our manuscript could benefit from the addition of TPB studies related to the COVID-19. As described in Response 2, we have included numerous recent studies in the introduction and discussion. Thank you again for your careful revision.

 

Reviewer #2: Congratulations on this well written piece of work. 

Response: We would like to thank Reviewer #2 for his/her encouraging feedback and useful comments.

Please find my comments below; "Several meta-analyses have provided evidence for the ability of the components of the TPB to predict general health behaviors (e.g., Conner & Norman, 2015). Interestingly, a few

studies have addressed these associations during the COVID-19 context, on several health behaviors such as physical distancing (Zhang et al., 2019), mask wearing (Chung et al., 2018) and intentions to receive a COVID-19 vaccine (Chu & Liu, 2021; Shmueli, 2021). However, the literature suffers from a lack of studies focusing on limitation of social contacts as a health behavior. This limitation is addressed in the present research."

- While that is true to an extent, you may want to have a look at this article published in PLOS1 that has assessed the behavior modifications during COVID-19 using the TPB on a wide multi-country scale (https://journals.plos.org/plosone/article?id=10.1371/journal.pone.0239961)

Response: This is a good suggestion. Specifically, we now cite Ammar et al. (2020)’s study in the introduction: “Interestingly, numerous studies have addressed these associations during the COVID-19 context, on several health behaviors such as physical distancing (Zhang et al., 2019), mask wearing (Kim et al., 2020), handwashing (Ammar et al., 2020), intentions of traveling (Hamid & Bano, 2021), and intentions to receive a COVID-19 vaccine (Chu & Liu, 2021; Shmueli, 2021).”

"suggest that the TPB model is a more universal rather than culture specific process in explaining protective behaviors, when comparing different targeted health behaviors among individuals living in different national contexts."

- I believe that may be an overstatement. Verily, Belgian and French nationals do differ appreciably with respect to culture and social norms, however, that difference is not striking, given their very close geographic location, european heritage and descent.

Response: This is a good point to consider. As suggested, we have tempered our conclusions and wrote that: “Importantly, the results of the present study indicate that (1) the four TPB components play a key role in developing and maintaining health behaviors during the COVID-19 crisis and (2) suggest that the TPB model tends to manifest similarly across countries in explaining protective behaviors, when comparing different targeted health behaviors among individuals living in different national contexts.” We have also made this change in the abstract.

- I am under the impression that one of the aims of this manuscript was to attest to the validity of the TPB, which has been extensively used and validated in the field of health research since its advent in 1991.

Response: With respect to Reviewer #2, the purpose of the present research was to compare the model of the TPB on handwashing and limitation of social contacts among people living in Belgium and France within the context of the COVID-19 pandemic. In order to compare our model within our two samples of French and Belgian participants, we had to report model fit statistics (e.g., CFI, RMSEA). Fortunately, these excellent model fit statistics simply demonstrate that the theoretical model holds perfectly in these two different national contexts, which allows us to compare and draw conclusions based on this cross-national study.

6. PLOS authors have the option to publish the peer review history of their article (what does this mean?). If published, this will include your full peer review and any attached files. Do you want your identity to be public for this peer review? For information about this choice, including consent withdrawal, please see our Privacy Policy.

Reviewer #1: Yes: Assoc. Prof. Yogi Tri Prasetyo, Ph.D.

Reviewer #2: No

Response: We would like to thank Associate Professor Yogi Tri Prasetyo and Reviewer #2 for their careful revision of our manuscript.

---

## [Decision Letter · Decision Letter 1]

24 Sep 2021

The theory of planned behavior during the COVID-19 pandemic: A comparison of health behaviors between Belgian and French residents

PONE-D-21-20498R1

Dear Dr. Wollast,

We’re pleased to inform you that your manuscript has been judged scientifically suitable for publication and will be formally accepted for publication once it meets all outstanding technical requirements.

Kind regards,

Camelia Delcea

Academic Editor

PLOS ONE

Additional Editor Comments (optional):

Reviewers' comments:

Reviewer's Responses to Questions

**Comments to the Author**

1. If the authors have adequately addressed your comments raised in a previous round of review and you feel that this manuscript is now acceptable for publication, you may indicate that here to bypass the “Comments to the Author” section, enter your conflict of interest statement in the “Confidential to Editor” section, and submit your "Accept" recommendation.

Reviewer #1: All comments have been addressed

Reviewer #2: All comments have been addressed

2. Is the manuscript technically sound, and do the data support the conclusions?

Reviewer #1: Yes

Reviewer #2: Yes

3. Has the statistical analysis been performed appropriately and rigorously? 

Reviewer #1: Yes

Reviewer #2: Yes

4. Have the authors made all data underlying the findings in their manuscript fully available?

Reviewer #1: Yes

Reviewer #2: Yes

5. Is the manuscript presented in an intelligible fashion and written in standard English?

Reviewer #1: Yes

Reviewer #2: Yes

6. Review Comments to the Author

Reviewer #1: The manuscript looks so much better now. Thank you for addressing all my comments. I think it is ready for publication. Congratulations.

Reviewer #2: Thank you for considering my previous comments. I have no further comments on your manuscript. Best wishes.

7. PLOS authors have the option to publish the peer review history of their article (what does this mean?). If published, this will include your full peer review and any attached files.

Reviewer #1: **Yes: **Assoc.Prof.Dr.Yogi Tri Prasetyo

Reviewer #2: No

---

## [Editor Report · Acceptance letter]

21 Oct 2021

PONE-D-21-20498R1 

The theory of planned behavior during the COVID-19 pandemic: A comparison of health behaviors between Belgian and French residents 

Dear Dr. Wollast:

I'm pleased to inform you that your manuscript has been deemed suitable for publication in PLOS ONE. Congratulations! Your manuscript is now with our production department. 

Kind regards, 

on behalf of

Dr. Camelia Delcea 

Academic Editor

PLOS ONE